# Fine Mapping and Candidate Gene Analysis of *Pm36*, a Wild Emmer-Derived Powdery Mildew Resistance Locus in Durum Wheat

**DOI:** 10.3390/ijms232113659

**Published:** 2022-11-07

**Authors:** Domenica Nigro, Antonio Blanco, Luciana Piarulli, Massimo Antonio Signorile, Pasqualina Colasuonno, Emanuela Blanco, Rosanna Simeone

**Affiliations:** 1Department of Soil, Plant and Food Sciences (DiSSPA), Genetics and Plant Breeding Section, University of Bari Aldo Moro, Via Amendola 165/A, 70126 Bari, Italy; 2Institute of Biosciences and Bioresources, National Research Council, Via Amendola 165/A, 70126 Bari, Italy

**Keywords:** *Blumeria graminis*, wheat, disease resistance, powdery mildew, backcross inbred lines, *Triticum turgidum* ssp. *dicoccoides*, wild emmer wheat, powdery mildew candidate genes

## Abstract

Powdery mildew (PM) is an economically important foliar disease of cultivated cereals worldwide. The cultivation of disease-resistant varieties is considered the most efficient, sustainable and economical strategy for disease management. The objectives of the current study were to fine map the chromosomal region harboring the wild emmer PM resistance locus *Pm36* and to identify candidate genes by exploiting the improved tetraploid wheat genomic resources. A set of backcross inbred lines (BILs) of durum wheat were genotyped with the SNP 25K chip array and comparison of the PM-resistant and susceptible lines defined a 1.5 cM region (physical interval of 1.08 Mb) harboring *Pm36*. The genetic map constructed with F_2:3_ progenies derived by crossing the PM resistant line 5BIL-42 and the durum parent Latino, restricted to 0.3 cM the genetic distance between *Pm36* and the SNP marker IWB22904 (physical distance 0.515 Mb). The distribution of the marker interval including *Pm36* in a tetraploid wheat collection indicated that the positive allele was largely present in the domesticated and wild emmer *Triticum turgidum* spp. *dicoccum* and ssp. *dicoccoides*. Ten high-confidence protein coding genes were identified in the *Pm36* region of the emmer, durum and bread wheat reference genomes, while three added genes showed no homologous in the emmer genome. The tightly linked markers can be used for marker-assisted selection in wheat breeding programs, and as starting point for the *Pm36* map-based cloning.

## 1. Introduction

Durum and common wheat are susceptible to a series of biotic adversities. Fungal diseases, such as rusts, Fusarium head blight, Helminthosporium leaf blight, Septoria leaf blotch, tan spot and powdery mildew, can cause substantial yield losses and alterations of grain quality properties [1]. Wheat powdery mildew, caused by *Blumeria graminis* (DC) Speer f. sp. *tritici* Em. Marchal (syn. *Erysiphe graminis* f. sp. *tritici*) (*Btg*), is one of the most economically important diseases in areas with high rainfall and semi-continental climate as well as in hot and dry climate regions where semi-dwarf wheat varieties, high nitrogen fertilization and irrigated practices are widespread [2,3]. Powdery mildew is an obligate biotrophic parasite as it depends on a living host for its growth and propagation [4]. The fungus lives on the surface of the host, infecting all parts of the plant (culm, sheath, leaves, awn, ear). Initially, the infected tissue shows diffuse chlorosis, then an epiphytic mycelium develops which can cover the entire leaf blade. The mycelium constitutes a barrier to photosynthetic activity and leads to an increase in transpiration; haustoria are responsible for delivering nutrients from the epidermal cells [5]. Wheat is susceptible during the stem elongation stage, but the most serious damage occurs in the heading-flowering stage. Infected plants lose vigor and grain yield suffers both quantitative and qualitative damage. Losses of 10–15% of grain yield can increase up to 40% when infection occurs before or during flowering; the recordable damage in the field varies from 13% to 34% with mild infestations and up to 50% with high disease severity [6].

The comprehensive prevention and control of wheat powdery mildew is based on the use of fungicides, the application of silicate, biological control and the cultivation of resistant varieties [7]. The growing interest in practices with higher environmental, economic and social sustainability has pushed the integration of the different methods of disease control with the aim of reducing the massive deployment of phytosanitary products, which can have an impact on the environment and the health of consumers and operators [8]. The cultivation of disease-resistant varieties is considered the most efficient, sustainable and economical strategy to prevent and control fungal diseases in wheat and other crops generally protected by chemical treatments [9]. Plant resistances to a pathogen species can be of qualitative and quantitative types [10]. The qualitative resistance (vertical resistance, oligogenic resistance, host-specific resistance) is controlled by a few major genes, generally of dominant inheritance, effective only against some races of the pathogen. Studies on the genetics of host–parasite interaction have led to the formulation of the gene-to-gene theory which involves the interaction of a resistance gene (*R*) of the plant and an avirulence gene (*Avr*) of the pathogen [11]. Recognition between the products of *R* and *Avr* genes triggers a series of events leading to the activation of defense mechanisms and the arrest of the pathogen growth [12]. So far, 68 loci for PM resistance have been identified (*Pm1-Pm68*) in the primary and secondary gene pools of cultivated wheat including *Triticum* and *Aegilops* species [13,14]. Some resistant genes have been also identified and transferred to wheat from the tertiary gene pool comprising some more distant species belonging to the genera *Secale*, *Thynopyrum* and *Dasypyrum* [13].

Thirteen PM resistance genes have been cloned: *Pm2* on chromosome arm 5DS [15], *Pm3* on 1AL [16], *Pm4* on 2AL [17], *Pm5* on 7BL [18], *Pm8* on 1BL [19], *Pm17* on 1RS.1AL [20], *Pm21* on 6VS.6AL [21,22,23], *Pm24* on 1DS [24], *Pm38* on 7DS [25], *Pm40* on 7BS [26], *Pm41* on 3BL [27], *Pm46* on 5DS [28] and *Pm60* on 7AL [29]. Most cloned genes encode a nucleotide binding and leucine-rich repeat (NBS-LRR) protein that activates effector-triggered immunity [30]. Exceptions are *Pm24* encoding a tandem kinase [24], *Pm4* and *Pm21* which encode a serine/threonine kinase [17,31] and *Pm26* and *Pm38* encoding a hexose transporter and an ABC transporter, respectively [25,28]. Another important exception is the *mlo* recessive gene, firstly studied in barley [32] and later reported in rice and wheat [33,34], which confers broad-spectrum resistance against each of the known *Bgt* isolates. Host-specific resistance confers a high degree of resistance to the host at seedling and adult plant, but it can be easily circumvented by the appearance of genetic variants of the pathogen characterized by new alleles at the *Avr* locus [13]. Consequently, new *R* genes must be continuously identified within the cultivated or wild species germplasm and used, through traditional approaches or advanced genetic biotechnologies, for varietal improvement [35,36].

The other type of resistance, often designated polygenic resistance, horizontal resistance, quantitative resistance or adult plant resistance (APR), is non-race specific and determined by several genes with additive effects (quantitative trait loci, QTL); often this type of resistance is partial and effective at adult-plant stage, but it is shown to be durable [37]. Genetic dissection of quantitative resistance is carried out on germplasm collections by genome-wide association mapping (GWAS) based on linkage disequilibrium (LD) [38], or on segregating biparental populations by the traditional linkage-based QTL mapping [39]. Both methodologies exploit the genetic association between molecular markers and QTL. Conventional mapping populations (F_2_, BC_1_, recombinant inbred lines, doubled haploids) have been generally used for mapping both QTL and major genes [40]. Advanced mapping populations, such as chromosome segment substitution lines, backcross inbred lines and near isogenic lines, are valuable genetic resources for basic and applied research on simple and complex traits which potentially address the limitations of conventional mapping populations 41. Advantages and disadvantages of the different segregating populations and their use in mapping studies were previously reported by Cavanagh et al. [41]. More than 100 QTL distributed on all wheat chromosomes have been detected and mapped in several studies; none of them have yet been cloned [13].

The screening of a set of 94 homozygous backcross inbred lines (BILs) for PM resistance, obtained by crossing the PM-susceptible durum wheat (*Triticum turgidum* ssp. *durum*) cv. Latino (recurrent parent) and the PM-resistant wild emmer wheat (*Triticum turgidum* ssp. *dicoccoides*) accession MG29896 (donor parent), identified three BILs (5BIL-29, 5BIL-42, 5BIL-50) displaying significant adult plant resistance to natural PM infection [42]. The phenotypic segregation pattern of two segregating populations (F_2:3_ progenies of the crosses 5BIL-29 x Latino and 5BIL-42 x Latino) evaluated at two-leaf stage seedlings and the molecular analysis with AFLP and SSR markers indicated a single dominant PM resistance gene on chromosome arm 5BL (bin 5BL6-0.55-0.76), later designated *Pm36* [42]. The main objectives of the current study were (a) to fine map the chromosomal region harboring the locus *Pm36* with SNP marker by using primary and secondary homozygote recombinant BILs and by linkage analysis in a segregating population; and (b) to identify candidate genes for *Pm36* by exploiting the improved tetraploid wheat genomic resources. Identification and characterization of genetic loci controlling seedling and adult plant resistance to powdery mildew will provide additional genetic resources to breeders to improve commercial cultivars of durum and bread wheat. The tightly linked markers to *Pm36* can be used in marker-assisted selection (MAS) programs and as starting point for the *Pm36* map-based cloning.

## 2. Results

### 2.1. Molecular Characterization of Backcross Inbred Lines

A set of 94 backcross inbred lines, obtained by backcrossing the PM resistant wild emmer wheat accession MG29896 with the PM susceptible durum wheat Latino (recurrent parent), were genotyped with the wheat SNP 15K chip array developed by Wang et al. [43] in order to characterize and define the introgressed dicoccoides 5BL chromosomal region carrying the *Pm36* locus, and possibly to dissect other useful quantitative traits in the future. The main features of the three PM-resistant lines (5BIL-29, 5BIL-42, 5BIL-50) and one relevant PM susceptible line (5BIL-20) are reported in Table 1 and Figure 1.

By using the durum consensus linkage map [44] as reference map, the introgressed 5BL dicoccoides chromosome segments carrying the *Pm36* locus were estimated to be 9.3 cM, 20.0 cM and 25.7 cM long in the PM-resistant lines 5BIL-42, 5BIL-50 and 5BIL-29, respectively. Their physical position was determined by BLAST-ing the 100 bp sequences including the SNP of the adjacent markers of each introgressed emmer segment against the durum wheat Svevo reference genome [45] and the wild emmer wheat Zavitan reference genome v2.0 [46]. The physical length ranged from 15.2 Mb in 5BIL-42, to 35.5 Mb in 5BIL-50, to 46.9 Mb in 5BIL-29. Comparison of the chromosome segments introgressed in the PM-resistant and -susceptible lines (Figure 1e) allowed us to define a 5BL target genetic region of 7.3 cM delimited by the centromeric proximal marker IWB22904 (cM 106.0) of the PM-resistant line 5BIL-42 and the centromeric distal marker IWB55478 (cM 113.3) of the PM-resistant line 5BIL-50.

### 2.2. Genetic Mapping of the Target Region Harboring Pm36

The PM-resistant line 5BIL-42 with the shortest introgressed *dicoccoides* segment was selected for further analysis and then crossed to the recurrent durum parent Latino to better define the *Pm36* position and to narrow down the introgressed region. A total of 252 F_2:3_ families were phenotyped at seedling stage for the response to the *Bgt* isolate O2 in a greenhouse experiment (Figure 2), and the observed phenotypic segregation ratio (75 homozygous resistant, 109 segregating and 68 homozygous susceptible) validated the hypothesis of the monofactorial control of the PM resistance (X2 = 4.58; 0.25 > *p* > 0.10).

These 252 F_2:3_ progenies, together with the 144 F_2:3_ progenies of the same cross previously developed [42], were genotyped with 18 SNP markers located in the target region (marker interval IWB22904-IWB55478) according to the durum wheat consensus map [44]. The EST-SSR BJ261635 marker, previously found to be tightly linked to *Pm36* [42], was also used to genotype all the 396 progenies. Two SNP markers failed and three had more than 10% of missing data and were discarded; 45 F_2:3_ progenies with more than 20% missing data were also removed. The resulting genetic map was then constructed with 351 progenies including 13 SNP markers, the EST-SSR BJ261635 marker and the *Pm36* phenotypic data (Figure 3). Surprisingly, *Pm36* mapped at centromeric proximal end of the genetic map tightly associated to the SNP markers IWB22904, IWB65455 and IWB69885 and to the EST-SSR marker BJ261635. The BLAST analysis of the BJ261635 and IWB22904 sequences against the Zavitan and Svevo reference genomes indicated that both markers were located within the same gene (*TRIDC5BG056780* of the emmer genome and *TRITD5Bv1G186920* of the durum genome). The total length of the genetic map was 10.0 cM, comparable to the length of the same chromosomal segment in the durum consensus map (9.3 cM) [44]. The physical-to-genetic distance ratio (1.71 Mb/cM and 1.74 Mb/cM in the durum and emmer genomes, respectively), and the estimated 0.3 cM genetic distance between *Pm36* and the nearest marker IWB22904, indicated that *Pm36* was in the centromeric proximal position at 515,731–520,760 bp from IWB22904.

Twelve 5BIL-42 x Latino F_2_:_3_ progenies heterozygotes for the *Pm36* target region were selected and selfed to the F_6_ generation to identify secondary recombinants with an introgressed emmer segment shorter than the 5BIL-42 one (Table 1). The genotyping of the relevant A15-F6 and PM222-F6 secondary recombinant lines with the wheat SNP 25K chip array allowed us to delimitate the chromosomal interval harboring *Pm36* between the polymorphic marker IWB22904 (cM 106.0) and the nearest non-polymorphic marker IWB7454 (cM 104.5) between 5BIL-42 and Latino. This genetic interval (1.5 cM) corresponds to a physical interval of 1,086,588 bp in the durum genome and 1,144,414 bp in the emmer wheat genome (Figure 3b).

### 2.3. Distribution of Pm36 in a Tetraploid Wheat Collection

The distribution of the IWB7454-IWB22904 marker interval including *Pm36*, potentially useful in durum wheat research and breeding programs, was investigated in a tetraploid wheat collection comprising 214 wild and cultivated accessions of seven *T. turgidum* subspecies (*durum*, *turanicum*, *polonicum*, *turgidum*, *carthlicum*, *dicoccum*, *dicoccoides)* previously evaluated at seedling and adult plants for PM resistance [47]. The positive allele was found in 20 (9.3%) accessions (Table 2), but its distribution varied greatly among subspecies: it was completely absent in ssp. *durum*, ssp. *turanicum*, ssp. *polonicum*, and ssp. *carthlicum*, and with a low frequency (*n* = 1, 6.3%) in the ssp. *turgidum*, while it was largely present in the domesticated and wild emmer spp. *dicoccum* and ssp. *dicoccoides* (66.7% and 77.8%, respectively). The low occurrence in cultivated or sporadically cultivated subspecies of *T. turgidum* could be attributed to the large use of well-known PM-resistance genes, such as *Pm2*, *Pm3*, *Pm4* and other genes derived from wheat landraces, in wheat breeding programs compared to the reduced use of the wild wheat resources for improving the PM disease resistance. However, not all accessions with the positive allele were PM-resistant, likely because the occurrence of recombination events between the molecular markers and the *Pm36* locus. Therefore, these markers should be investigated in the specific materials before being used for marker-assisted selection in wheat breeding programs.

### 2.4. Candidate Genes of Pm36

The sequences of the markers IWB7454 and IWB22904, delimiting the 5BL genomic region including *Pm36* within a genetic interval of 1.5 cM, were used to define the corresponding physical interval in the wild emmer wheat Zavitan v2.0 reference genome [46], in the durum wheat Svevo reference genome [45] and in the bread wheat Chinese Spring v1.0 reference genome [48]. The target regions, corresponding to 1,144,414 nucleotides (555,463,725–556,608,139) in the Zavitan sequence, to 1,086,588 nucleotides (537,358,372–538,445,060) in the Svevo sequence and to 1,076,901 nucleotides (540,266,345–541,343,246) in the Chinese Spring genome (https://wheat.pw.usda.gov) (10 October 2022), were further investigated to identify candidate genes involved in the PM resistance.

Thirteen high-confidence protein-coding genes were found to be annotated (Figure 4, Appendix A), including the two genes within which the sequences of the two markers IWB7454 and IWB22904 flanking the target interval were found. A highly micro-collinearity was detected among the wild emmer wheat and the cultivated durum and bread wheat. Three genes identified in durum and bread wheat showed no homologue in the *dicoccoides* genome: *TRITD5Bv1G186610*, encoding a F-box family protein, *TRITD5Bv1G186630*, encoding a Transmembrane protein, putative (DUF594), and *TRITD5Bv1G186820* encoding a Ulp1 protease family, C-terminal catalytic domain containing protein. Out of the remaining ten genes underlying the target region, six were found to be encoding for BTB/POZ domain-containing family protein, and the other four being a Mitochondrial glycoprotein, a Cleavage and polyadenylation specificity factor subunit 3, a Purine permease-related family protein and a Cytochrome b561 and domon domain-containing protein.

The WheatOmics 1.0 database (http://wheatomics.sdau.edu.cn/) (10 October 2022) was searched by querying the homologous Chinese Spring gene IDs to carry out in silico expression analysis aimed to determine whether a differential expression after powdery mildew inoculation was shown (expression data on leaves sampled 24, 48 and 72 h after inoculation [49]). Interestingly, five genes showed a significant differential expression. *TraesCS5B02G360900*, a homologous gene of *TRITD5Bv1G186620*, encoding for a Cleavage and polyadenylation specificity factor subunit 3, was found to be significantly differentially expressed at 48 and 72 h after inoculation compared to control. *TraesCS5B02G361100*, a homologous gene of *TRITD5Bv1G186700*, encoding for a Purine permease-related family protein, was found to be significantly differentially expressed at 72 h after inoculation. The other three genes all encoded for a BTB/POZ domain-containing family protein; *TraesCS5B02G361200* and *TraesCS5B02G361500*, homologous of *TRITD5Bv1G186770* and *TRITD5Bv1G186830*, respectively, were found to be significantly differentially expressed at 48 and 72 h after inoculation, while *TraesCS5B01G361700*, a homologous gene of *TRITD5Bv1G186880*, was differentially expressed only at 72 h after inoculation. Expression data of the above-mentioned genes are reported in Appendix A.

## 3. Discussion

### 3.1. Comparison of Pm36 with Known Powdery Mildew Resistance Genes and QTL Mapped on 5BL

Seven PM loci, *Pm36* [42], *MlWE29* [50], *M3D232* [51], *PmAS846* [52], *MlWE4* [53], *PmG25* [54] and *Pm53* [55], were located on chromosome arm 5BL by using different SSR, EST-SSR, EST-STS and SNP markers. These loci were all found to be dominant or incompletely dominant PM-resistant genes. Comparative mapping was carried out by the comprehensive wheat genomic resources recently developed in tetraploid wheat, such as the high-density durum genetic linkage map [44], the wild emmer Zavitan reference genome [46], the durum Svevo reference genome [45], and the web-based tools providing wheat genomic data (https://plants.ensembl.org/index.html; https://wheat.pw.usda.gov/) (10 October 2022). Sequences of the different types of markers employed in the above PM mapping studies were used in BLAST analyses to determine the physical position of adjacent markers of each *Pm* locus in the wild emmer and durum references genomes (Figure 5).

*Pm36*, *PmAS846* and *MlWE4*, all derived from wild emmer accessions, covered the physical interval 554.27–556.61 Mb in the emmer genome and 536.38–538.44 Mb in the durum genome. These three PM loci, tightly associated to the EST-SSR markers BJ261635 and BD37680 and to the SNP marker IBW22904, whose sequences were found to be located within the same gene (*TRITD5Bv1G186920*), very likely correspond to different alleles of the same PM locus or to different loci located in a short 5BL genomic region.

The flanking markers of the PM locus *M3D232*, originated from the wild emmer accession I222 [51], cover the physical chromosomal interval 530.73–540.86 Mb in the emmer genome and 515.39–524.34 Mb in the Svevo genome, and therefore should be a PM-resistant gene different from *Pm36*, *PmAS846* and *MlWE4*.

The markers *wmc415* and *wmc289*, flanking *MlW29*, which originated from the emmer accession WE29 [50], the markers *gpw3191* and *FCP1* flanking *PmG25*, which originated from the emmer accession G25 [54] and the SNP markers IWA2454 and IVA6024 adjacent to the *Pm53*, which were introgressed from *Aegilops speltoides* into common wheat [55], cover a 5BL genomic interval too large that does not allow the clarification of the exact relationships among them and with the other mentioned PM loci *Pm36*, *PmAS846 and MlWE4.*

Some QTL for PM resistance have been also detected on the long arm of chromosome 5BL: *QPm.sfr-5B* [56], *QPm.inra-5B.2* [57], *Qpm.caas-5BL.1*, *Qpm.caas-5BL.2*, *Qpm.caas-5BL.3* [58]. The QTL positions established by RFLP and SSR markers reported in the above investigations are not comparable and do not allow us to determine the relationships with the other mentioned PM loci. High-density marker maps or allelism tests could clarify their relationships.

### 3.2. Candidate Genes and In Silico Expression

The recently released and publicly available wild emmer [46] and durum wheat genomes [45] represent great tools in unveiling the functional role of potential candidate genes involved in specific metabolic pathways as well as for investigating surrounded genomic region. The analysis of the physical region containing the *Pm36* locus investigated in this work showed the presence of ten and thirteen high-confidence genes in wild emmer and durum wheat genomes, respectively, including F-box family protein, Cleavage and polyadenylation specificity factor subunit 3, Purine permease-related family protein, and several BTB/POZ domain-containing family proteins.

F-box proteins have regulatory function in protein degradation through E3 ubiquitin ligase proteolytic mechanism in response to several cellular signals during plant development and growth, as well as hormone and biotic/abiotic stress responses [59]. A recent paper [60] focused on analysing this protein family in the wheat genome along with their expression profiling at different developmental stages. Almost 1800 F-box genes were identified, and GO annotation assigned to more than 1000 of them. Ubiquitination and ubiquitin activity proteins were the most represented. Several papers found this class of proteins also involved in plant response to both biotic/abiotic stresses. In wheat, the cyclin F-box domain *TaCFBD* gene is involved in inflorescence development and cold stress responses [61]. The TaFBA1 F-box protein with an FBA domain has been implicated in drought and heat stress tolerance [62], whereas the F-box protein TaJAZ1 regulates resistance against powdery mildew [63].

Cleavage and polyadenylation of precursor mRNA is a crucial process for mRNA maturation. A subgroup of the protein known as “cleavage and polyadenylation specificity factor (CPSF)” is needed for both cleavage and polyadenylation in plants and animals. Proteomic studies carried out by Zhao et al., [64] characterized each subunit along with other polyadenylation related proteins. Cleavage and polyadenylation specificity factor subunit 3 (CPSF3, also known as CSFP73-I) is one of the components of CPSF complex in plants, which play a key role in pre-mRNA 3′-end formation but may also function as mRNA 3′-end-processing endonuclease and be involved in the histone 3′-end pre-mRNA processing [65].

Polyadenylation regulation of gene expression by CPSF30 occurs in a subset of biological processes, including plant development, disease resistance, and abiotic stress tolerance [66]. To date, the 73Kda subunit of CPSF complex has been extensively studied in *Arabidopsis*, where it has been demonstrated to affect reproductive development, especially early embryo development [67]. The significant expression after powdery mildew inoculation compared to control in bread wheat (Appendix A) suggests that a further and more focused analysis of this gene might be considered to confirm its eventual involvement in biotic stress response.

Purine permeases (PUPs) have been extensively studied in Arabidopsis. Among the 21 genes identified within AtPUP gene family, three members have been comprehensively studied—AtPUP1, AtPUP2 and AtPUP3 [68]—and their function established. AtPUP1 and 2 mediate energy-dependent high-affinity adenine uptake and cytokinin transport while AtPUP3 influences the transport of purine and purine derivatives during pollen tube elongation and pollen germination [69]. The AtPUP family not only mediated the transport of adenine and CKs but also mediated the transport of other compounds such as adenine, cytosine or secondary compounds such as cytokinins and caffeine [70]. In silico expression data showed a bread wheat purine permease located on 5B chromosome being significantly expressed at 72 h after powdery mildew inoculation, thus suggesting the potential involvement of other PUP family members in biotic stress resistance (Appendix A).

An interesting outcome from our study was the detection of six BTB/POZ domain-containing family proteins within the physical interval including *Pm36*, and three of them were found to be significantly expressed in bread wheat (Appendix A). The sequence and structural analysis of this domain proteins was reported by Stogios et al. [71]. The BTB domain, bric-à-brac, tramtrack and broad complex transcription regulators (also known as the POZ domain, poxvirus and zinc finger) is a protein–protein interaction motif in eukaryotes. The BTB/POZ domain has a highly conserved region and is often present at the protein N terminus. The wide range of sequence and length variation (approximatively 120 amino acids) between orthologs and paralogs suggested their potential involvement in different molecular, biochemical and biological functions [71]. Indeed, a variety of functional roles were firstly identified for the domain such as transcription repression [72] and protein ubiquitination/degradation [73]. More recent studies identified the BTB/POZ protein family involved in plant growth and development as well as in plant defense regulation in responses to biotic and abiotic stresses. Genome-wide analysis of the BTB/POZ gene family in different plant species confirmed some members being involved in the regulation of gametophyte development, seed germination, inflorescence architecture and branching [74]. Interestingly, others have been found to be engaged in environmental responses, including defense against pathogens and parasites, as well as responses to nutrient shortage, toxic heavy metals, and several other stressors [75,76].

A further analysis of the candidate genes we identified will be necessary to confirm their involvement in the regulation of plant responses to environmental stresses and eventually exploit their applications in stress tolerance engineering.

### 3.3. Pm36 Value in Wheat Breeding

The genetic resistance of cultivated wheat germplasm can be overcome due to the emergence of new *Bgt* races capable of generating severe PM epidemics [77,78]. The identification and validation of novel genes/QTL for PM resistance can successfully contribute to enrich the resistance source available to wheat breeders. The wild emmer wheat, a progenitor of cultivated durum and bread wheat, is a rich genetic resource of favorable genes useful for wheat improvement including resistance genes for fungal diseases (stripe rust, leaf rust, stem rust, powdery mildew, Fusarium head blight) [79]. At least 20 PM-resistance genes have been so far identified and characterized in the wild emmer wheat [80]. This wild wheat can be easily crossed with the cultivated common and durum wheat, its A and B genome chromosomes readily pair with their wheat homologues and recombinants can be selected in segregating populations without linkage drag problems [79]. Thus, the rich genetic diversity of this wild wheat can be usefully used to diversify the source of *Bgt* resistance genes in cultivated wheat. In the current work, a short chromosomal region of the wild emmer wheat, delimited by the SNP molecular markers IWB7454 and IWB22904 and the EST-SSR BJ261635 and including the PM-resistant *Pm36* locus, was transferred to a durum wheat cultivated background. Fine mapping using homozygous backcross inbreed lines and a segregating population delimited *Pm36* to a physical interval of approximately 1086 kb region in the durum genome containing thirteen annotated high-confidence protein-coding genes.

These results provide valuable information for marker-assisted selection (MAS) for powdery mildew resistance to expedite the selection of wheat cultivars with improved disease resistance and represent a starting point for the map-based cloning of the *Pm36* gene.

## 4. Materials and Methods

### 4.1. Plant Materials

A set of 94 backcross inbred lines (BILs) was previously developed from crossing one PM resistant accession (MG29896) of the wild emmer wheat (*T. turgidum* ssp. *dicoccoides)*, with the semi-dwarf and high-yielding durum wheat cv. Latino as described by Blanco [42]. Briefly, 110 F_1_ plants were backcrossed five times to the durum recurrent parent to achieve BC_5_F_1_ plants, which were selfed to the BC_5_F_7_ generation using the single-seed descent method. No selection was carried out during the backcrossing and self-fertilization generations; however, some lines were lost and the BC_5_F_7_ generation consisted of 94 BILs. In the current study the BILs population and the parental lines were genotyped with the wheat SNP 15K chip array developed by Wang et al. [43]. A segregant population of 252 F_2_-F_3_ progenies was developed by crossing the PM-resistant line 5BIL-42 with the durum wheat Latino and tested for the PM resistance at seedlings at two-leaf stage in a greenhouse experiment. A set of secondary recombinant lines was produced by selfing 12 selected F_3_ progenies heterozygote for the *Pm36* target region to the F_6_ generation.

### 4.2. Powdery Mildew Resistance Assays

The PM evaluation of 252 F_2_-F_3_ progenies and the parental lines 5BIL-42 and Latino was carried out under controlled greenhouse conditions with the highly virulent *Bgt* isolate O2 [49]. A total of 20 seeds of each progeny and parent were sown in 15 cm-diameter round pots and grown to the two-leaf stage. A plastic tower was used for the inoculation of PM spores at a density of 4 × 10^3^ conidia cm^−2^. When the susceptible durum parental line Latino showed fully developed PM symptoms (12 days after inoculation), each F_2_-F_3_ progeny was individually assessed as PM homozygous resistant, segregating and homozygous susceptible.

### 4.3. DNA Extraction and SNP Marker Analysis

The GeneElute Plant Genomic Miniprep Kit (Sigma, Waltham, MA, USA) was used for DNA extraction from fresh leaves of each BIL, parental line and F_2_:F_3_ progeny. DNA concentration and quality were checked by both agarose gel-electrophoresis and NanoDrop2000 (Thermo Scientific™, Waltham, MA, USA). Genomic DNA of each sample was diluted to 50 ng/µL and sent to TraitGenetics GmbH (Gatersleben, Germany) (http://www.traitgenetics.de) for the sample genotyping with the wheat SNP 15K or 25K chip array developed by Illumina CSPro^R^ (San Diego, CA, USA) as described by Wang [43].

Based on genomic locations of SNP markers in the Zavitan reference genome, Svevo reference genome and the durum consensus map [44,45,46], 18 SNP markers (IWA4793, IWB10356, IWB1762, IWB22904, IWB28447, IWB35312, IWB30236, IWB35880, IWB44719, IWB5145, IWB51987, IWB55478, IWB60173, IWB65455, IWB66419, IWB67424, IWB69885, IWB72546) were identified in the 5BL chromosome region including *Pm36* to be used in linkage analysis. PCR-based KASP (Kompetitive Allele-Specific PCR) genotyping of each progeny of the segregant F_2_:F_3_ population was performed by LGC Biosearch Technologies, Herts, UK (servicelab-uk@lgcgroup.com). The BJ261635 EST–SSR marker was amplified by the primer pair F(5′ TAGCCTGGTACCATTCTGCC) and R(5′ CTAGGCCTTCTGGTGTAATG) as described by [42]. The amplification products were separated by capillary electrophoresis in an automated DNA sequencer (ABIPRISM 3100 Avant, Applera, Norwalk, CT, USA).

### 4.4. Linkage Analysis

Goodness-of-fit at *p* > 0.01 of segregation ratios to expected ratio for each marker and PM segregation data was determined by the Chi-squared test. Markers with more than 10% missing data and F_2:3_ progenies with more than 20% missing data were removed from linkage analysis. Linkage between markers and determination of the linear order of loci was performed by QTL IciMapping 4.2. The Kosambi mapping function was used to calculate map distances.

## Figures and Tables

**Figure 1 ijms-23-13659-f001:**
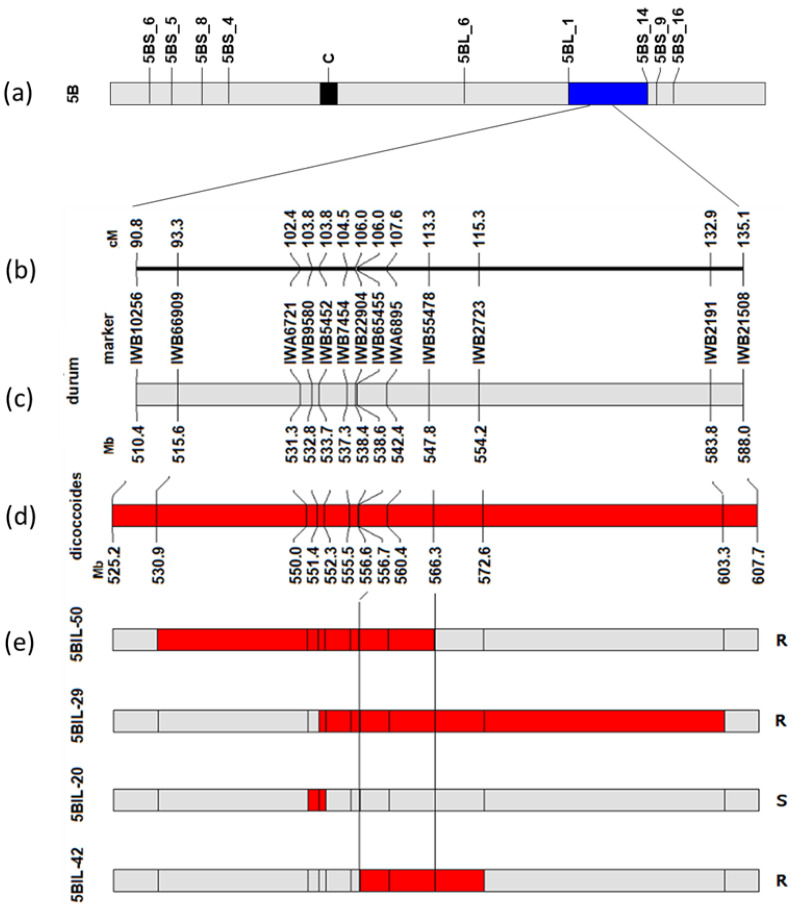
Fine mapping of the wild emmer-derived powdery mildew resistance *Pm36* locus using homozygous backcross inbred lines (BILs) of durum wheat. (**a**) Physical location of *Pm36* on the 5BL chromosome arm (bin 5BL_1) (blue block); (**b**) Genetic map of the 5BL region harboring *Pm36* with the genetic position (cM) of relevant SNP markers according to the consensus durum map [44]; (**c**,**d**) Schematic representation of the durum wheat Svevo reference genome (grey block) [45] and the *dicoccoides* Zavitan reference genome v2.0 (red block) [46] with the physical position (Mb) of relevant SNP markers; (**e**) Genotype and phenotype of four BILs (5BIL-50, 5BIL-29, 5BIL-20, 5BIL-42). S and R on the right of the bars indicate susceptible and resistant phenotype, respectively. Gray and red rectangles represent the genotypes of the PM-susceptible durum wheat Latino and the PM-resistant lines, respectively. *Pm36* was placed within a 7.3 cM interval flanked by IWB55478 and IWB22904 (between two vertical lines) based on the genotypes of the BILs at the marker loci and their PM phenotypes.

**Figure 2 ijms-23-13659-f002:**
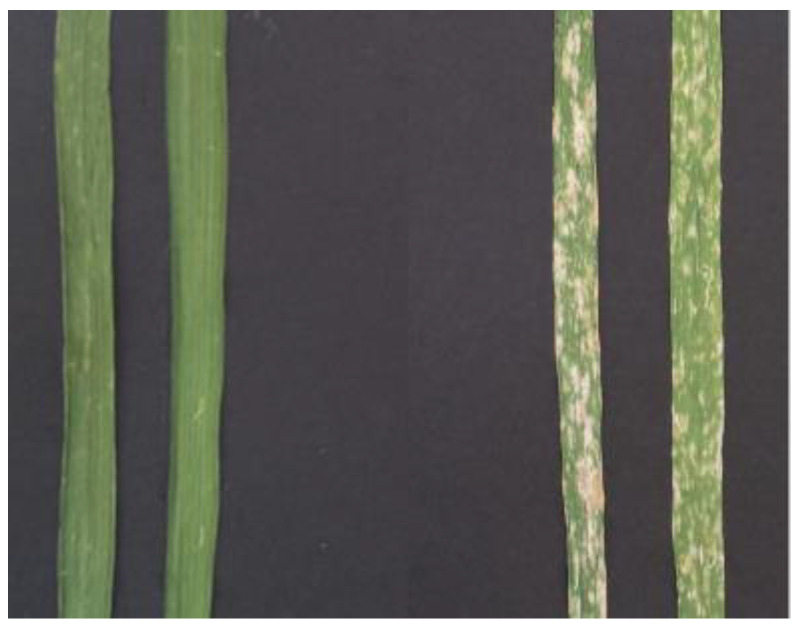
Phenotypes of powdery mildew resistant durum backcross inbred line 5BIL-42 (on the **left**) and of the durum recurrent parental line Latino (on the **right**) 12 days post-inoculation with *Bgt* race O2.

**Figure 3 ijms-23-13659-f003:**
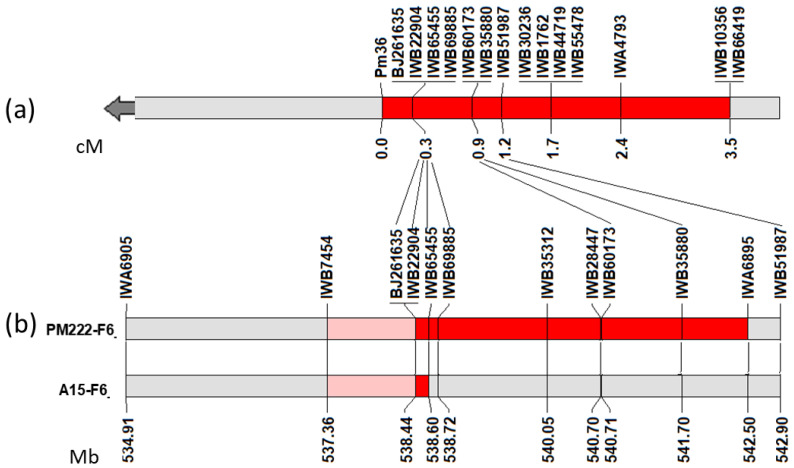
Fine mapping of *Pm36*. (**a**) Genetic linkage map of the 5BL chromosome region harboring *Pm36* generated in the 5BIL-42 x Latino F_2:3_ progeny population. The grey and red regions represent the durum and the *dicoccoides* chromosomal segments, respectively. Marker loci are listed above and genetic distances in cM are shown at the bottom of the chromosome bar. The arrow point to the centromere; (**b**) Secondary recombinant homozygous lines (A15-F6 and PM222-F6) including introgressed *dicoccoides* chromosomal regions shorter than the 5BIL-42 one; the pink segments indicate the chromosomal interval harboring *Pm36*. Marker loci are listed above and physical distances in Mb are shown at the bottom of the chromosome bar. The lines connect the common markers between the genetic map and the physical map of the secondary recombinant lines.

**Figure 4 ijms-23-13659-f004:**
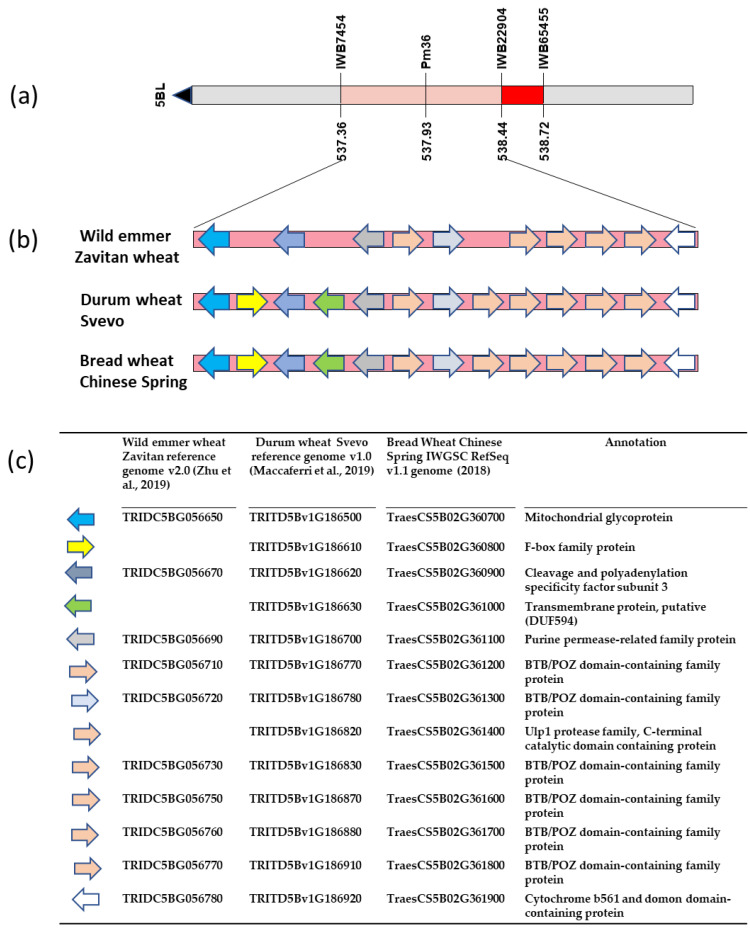
Fine mapping and candidate genes of *Pm36*. (**a**) High-density map positioned *Pm36* within a 1.08 Mb physical interval in the durum wheat Svevo reference genome [45]. Black arrow indicates direction of the centromere; (**b**) Micro-collinearity of the genomic region of *Pm36* between the wild emmer Zavitan [46], durum wheat Svevo [45] and bread wheat Chinese Spring genomes [48]. Arrows represent the annotated genes in each species and their direction indicates which strand they are located on; (**c**) Genes’ ID and their annotated functions in the emmer, durum and bread wheat genomes.

**Figure 5 ijms-23-13659-f005:**
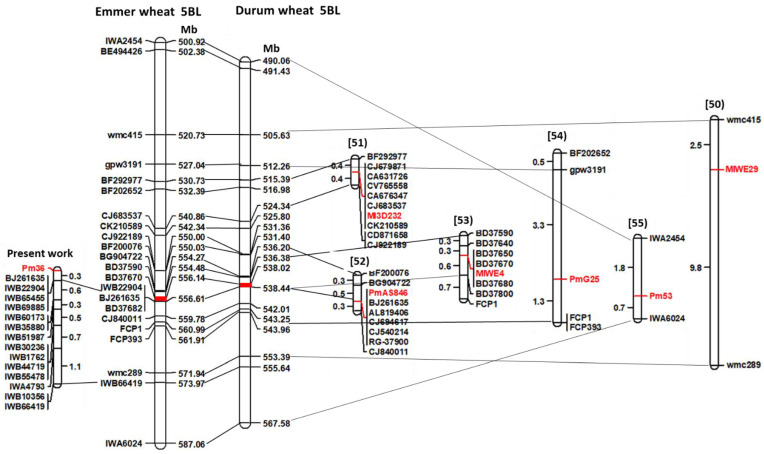
Comparison of the genetic linkage map of *Pm36* with those reported for the powdery mildew resistance genes *MlWE29* [50], *M3D232* [51], *PmAS846* [52], *MlWE4* [53], *PmG25* [54], *Pm53* [55] located on the 5BL chromosome arm. Marker sequences of each map were physically located (in Mb) on the wild emmer Zavitan reference genome v2.0 [46] and on the Svevo reference genome v1.0 [45] to make possible the comparison of genetic maps constructed with different molecular markers. Marker loci are listed to the left of the emmer and durum physical maps and Mb distances are shown to the right. The partial comparison maps show the references above the maps, the marker loci on the right and the genetic distance in centiMorgan (cM) on the left. The lines connect the flanking markers of each map with their physical location (in Mb) on the emmer and durum genomes. The red blocks indicate the *Pm36* genomic interval.

**Table 1 ijms-23-13659-t001:** Genetic and physical position in the durum wheat Svevo reference genome and in the wild emmer wheat Zavitan reference genome of the 5BL introgressed *dicoccoides* segments in six backcross inbred lines of durum wheat.

Lines	Adjacent SNP Markers	Genetic Position *	Genetic Length *	Durum Wheat Svevo Reference Genome v1.0		Wild Emmer Wheat Zavitan Reference Genome v2.0		Powdery Mildew
		(cM)	(cM)	Physical Position (bp)	Length (bp)	Physical Position (bp)	Length (bp)	
5BIL-20	IWB6721-IWB5452	102.4–103.8	1.4	531,347,607–533,723,310	2,375,703	549,987,942–552,272,976	2,285,034	Susceptible
5BIL-50	IWB66909-IWB55478	93.3–113.3	20.0	515,585,430–547,845,670	32,260,240	530,853,710–566,309,836	35,456,126	Resistant
5BIL-29	IWB9580-IWB2191	103.8–132.9	29.1	532,842,874–583,808,641	50,965,767	551,389,765–603,305,150	51,915,385	Resistant
5BIL-42	IWB22904-IWB2753	106.0–115.3	9.3	538,444,960–55,4157,612	15,712,652	556,608,139–572,559,558	15,951,419	Resistant
Secondary recombinants								
PM222-F6	IWB22904-IWA6895	106–107.6	1.6	538,444,960–542,499,644	4,054,684	556,608,139–560,376,739	3,768,600	Resistant
A15-F6	IWB22904-IWB65455	106.0	0.0	538,444,960–538,604,270	159,310	556,608,139–556,748,732	140,593	Resistant

* Genetic position in the durum wheat consensus map [44].

**Table 2 ijms-23-13659-t002:** Frequency of the Pm36 resistant allele (chromosome regions included by the flanking markers IWB7454 and IWB22904) in a collection of 214 tetraploid wheat accessions evaluated for their reaction to *Blumeria graminis f*. sp. tritici at adult plant and seedling (race O2) stages. All lines were classified into two groups with resistant lines scoring 0–2 at seedlings and adult plants, and susceptible lines scoring 2.1–4.0 at seedlings and 2.1–9.0 at adult plants [47].

*T. turgidum* Subspecies	Total Number of Accessions	Frequency of Resistant Allele		Adult Plants (Scale 0–9)		Seedling (Scale 0–4)	
				Resistant Accessions		Resistant Accessions	
		N.	%	N.	%	N.	%
*durum*	121	0		0		0	
*turanicum*	20	0		0		0	
*polonicum*	19	0		0		0	
*turgidum*	16	1	6.3	0		0	
*carthlicum*	11	0		0		0	
*dicoccum*	18	12	66.7	11	91.7	5	41.7
*dicoccoides*	9	7	77.8	6	85.7	6	85.7
Whole collection	214	20	9.3	21	105.0	16	80.0

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
