# Peer review of "Fine Mapping and Candidate Gene Analysis of Pm36, a Wild Emmer-Derived Powdery Mildew Resistance Locus in Durum Wheat"

_ijms, 2022, doi:10.3390/ijms232113659_

Round 1
Reviewer 1 Report
This is a useful study addressing a wild emmer-derived powdery mildew resistance locus in durum wheat. Authors should check the following lines for English and clarity:
line 25 (tightly linked what?), 29 (please give the genus name of dicoccoides also), 33, 39, 45, 55, 56, 69, 108 (please write the scientific name of wild emmer), 107 (please write the scientific name of durum wheat), 153, 266-267, 282, 289, 298, 305, 342, 359-360 (Arabidopsis should be italic), 365, 367, 370, 404, 413, 414, 432, 457 ( as described by....(43), 467 (as described ...(42), 444-445 (please give details of the inoculation process), 523, 568, 615, 621, 651, 654, 655-656, 670
Author Response
Many thanks for your comment and suggestions.
- Line 25: “Markers” was added to the sentence: “The tightly linked markers can be used for marker-assisted selection ………. “.
- Lines 25 and 29: The species name has been reported: Triticum turgidum ssp. dicoccoides.
- Line 107: The species name has been reported: Triticum turgidum ssp. durum.
- Line 359-360: Arabidopsis has been reported in italic: Arabidopsis.
- Line 457: revisioned “….. as described by Blanco [42].”
- Line 467: revisioned “….. as described by Wang [43].”
- Lines 444-445 (please give details of the inoculation process): Details of the inoculation process are reported in lines 440-447.
Reviewer 2 Report
"Fine mapping and candidate gene analysis of Pm36, a wild emmer-derived powdery mildew resistance locus in durum wheat" is a fine effort to limit the QTL region of the of Pm36 to assist MAS in wheat. As such, PM is a devastating disease and its impact is quite severe in the humid Mediterranean climate of Central and also colder climate of northern Europe. Hence, the outcome of this article will appear in the form of varietal development using the MAS in breeding programs.
I have made few comments in the attached file for your reference which need to be addressed.
Best regards

Author Response
Many thanks for your comment and suggestions.
Accordingly, the text has been revised as follows:
- Line 25: “Markers” was added to the sentence: “The tightly linked markers can be used for marker-assisted selection ………. “.
- Line 60: “species” was deleted.
- Line 84: “appearance” was substituted by “evolution”.
- Line 98: the reference was added.
- Line 221: Pm36 has been always used in italic in the whole text.
- Lines 260 and 372: “in silico” has been used in italic.
- Line 261: “powdery mildew” has been substituted by PM.
- Line 282: “PM resistant genes” has been deleted.
- Lines 419-421.
As reported in the section “Distribution of Pm36 in a tetraploid wheat collection”, Pm36 was completely absent in cultivated subspecies of T. turgidum, but it was largely present in the wild emmer wheat.
According to the comment, the following sentences was added in the section “Distribution of Pm36 in a tetraploid wheat collection” to explain the low occurrence of Pm36 in cultivated wheat.
“The low occurrence in cultivated or sporadically cultivated subspecies of T. turgidum could be attributed to the large use of well-known PM resistance genes, such as Pm2, Pm3, Pm4 and other genes derived from wheat landraces, in wheat breeding programs compared to the reduced use of the wild wheat resources for improving the PM disease resistance”.